# Determination of the Minimum Detectable Change in the Total and Segmental Volumes of the Upper Limb, Evaluated by Perimeter Measurements

**DOI:** 10.3390/healthcare8030285

**Published:** 2020-08-21

**Authors:** José Manuel Tánori-Tapia, Ena Monserrat Romero-Pérez, Néstor Antonio Camberos, Mario A. Horta-Gim, Gabriel Núñez-Othón, Carlos Medina-Pérez, José Antonio de Paz

**Affiliations:** 1Division of Biological Sciences and Health, University of Sonora, Hermosillo 83000, Mexico; josemanuel.tanori@unison.mx (J.M.T.-T.); nestor87pb@hotmail.com (N.A.C.); gabriel@guaymas.uson.mx (G.N.-O.); japazf@unileon.es (J.A.d.P.); 2Sciences Health School, University Isabel I, 09003 Burgos, Spain; carlosmedinaper85@gmail.com; 3Institute of Biomedicine, University of León, 24071 León, Spain

**Keywords:** breast cancer, lymphedema, minimal detectable change, standard error of measurement, reliability, circumference measurement

## Abstract

Among female breast cancer survivors, there is a high prevalence of lymphedema subsequent to axillary lymph node dissection and axillary radiation therapy. There are many methodologies available for the screening, diagnosis and follow-up of breast cancer survivors with or without lymphedema, the most common of which is the measurement of patients’ arm circumference. The purpose of this study was to determine the intra-rater minimal detectable change (MDC) in the volume of the upper limb, both segmentally and globally, using circumference measurements for the evaluation of upper limb volume. In this study, 25 women who had received a unilateral mastectomy for breast cancer stage II or III participated. On two occasions separated by 15 min, the same researcher determined 11 perimeters for each arm at 4 cm intervals from the distal crease of the wrist in the direction of the armpit. The MDC at the segmental level ranged from 3.37% to 7.57% (2.7 to 14.6 mL, respectively) and was 2.39% (42.9 mL) at the global level of the arm; thus, minor changes in this value result in a high level of uncertainty in the interpretation of the results associated with the diagnosis of lymphedema and follow-up for presenting patients.

## 1. Introduction

Statistics provided by the Global Cancer Observatory [1] estimate that 18,078,957 people worldwide were diagnosed with cancer in 2018. Of these, 8,622,539 were women and 2,088,849 had breast cancer (BC), meaning that BC accounted for 24.2% of all new cancers in women in 2018 [2].

From 1975 to 2000, the incidence of BC increased each year by 0.5–1.5%, varying between countries [3], and between 2005 and 2015, this escalated to approximately 33% [4]. While the global mortality rate continues to increase, primarily at the expense of populations in poorer regions, the 5- and 10-year survival rates are approximately 83% and 72%, respectively. However, there are large racial differences; for example, within the same country, the 5-year survival rate was observed to be 81% for black women and 92% for white women [5].

Women who have suffered from BC often present short- or long-term sequelae, such as psychological sequelae (i.e., depression, anxiety, cognitive impairment, body image disorders, or sexual dysfunction), fatigue, pain in the operated area, polyarticular pain, decline in physical condition, or lymphedema [6].

Lymphedema is a chronic, potentially progressive disorder which is characterized by the pathological accumulation of lymphatic fluid and fibrosis of subcutaneous tissue somewhere in the body. In BC survivors, this tissue is in the ipsilateral arm of BC and most often appears after axillary lymph node dissection and axillary radiotherapy [7,8]. This secondary lymphedema often causes functional problems, adversely affects quality of life, and may be accompanied by recurrent infections [9]. However, the published prevalence of lymphedema in women who have suffered from BC varies significantly, with a 5-year prevalence of 42% [10] and a 2-year prevalence of 5% [11]. Moreover, globally, it is estimated that approximately 20% of women who survive BC will develop arm lymphedema [12].

There are many methodologies available for the screening, diagnosis, and follow-up of BC survivors with or without lymphedema, namely, water displacement, perimetry, circumference measurements, bioimpedance spectroscopy, lymphoscintigraphy, imaging tests such as ultrasound and magnetic resonance imaging, or self-reporting. However, circumference measurements are most commonly used by healthcare providers who follow up these patients [13], and they may be used either as raw scores or be converted to a volumetric measurement of the intervening segment using a geometric formula. They are used in the follow-up of BC survivors in terms of screening, early diagnosis, and follow-up of lymphedema. In a meta-analysis, data from 83 studies were collated, in which different methods were used for the evaluation of lymphedema [12]; circumference measurement was found to be the most frequently used method (38 studies), followed by questionnaires (19), perimetry (17), and bioelectric impedance analysis (3). In another more recent meta-analysis that included the results of five studies, circumference measurement was also found to be the most frequently used method (33 studies), followed by water displacement (20) and perimetry (7). Other methods were also noted, but they were even less frequently used [14].

Perimeter measurements are known to be consistent with other methods, such as water displacement, perimetry, and resonance, generally showing good levels of correlation and good repeatability, especially when conducted with a narrow-bladed tape [15].

Swelling is not always widespread throughout the limb, and instead occasionally appears in only a limited segment, making it difficult to use methods that assess the entire limb, such as water displacement. Circumference measurement is a simple, fast, and cheap technique. Referral to a medical service specializing in lymphedema is recommended when differences of ≥10% are detected between the extremities [16], although, depending on the difference between limbs, many clinicians prefer to rate the lymphedema as minimal (>5–10%) or mild (>10% to <20%) [17].

For all of these reasons, it is important to have knowledge and a good understanding of the precision and degree of uncertainty of the measurements as well as the sensitivity of this instrument, in order to detect changes and to be able to draw better conclusions from its use in the follow-up of patients with lymphedema. It is often assumed that a patient with lymphedema has improved when, after two separate measurements, the second shows a lower value. However, to determine whether an improvement is real, it is necessary to know the minimum change values that can be detected by the used instrument (i.e., the minimal detectable change (MDC)). If the difference between the evaluations is greater than the MDC value of the employed measurement method, one can be sure, with a high degree of certainty, that the variation observed is not due to a limitation or random error in the method used for measurement.

In the literature, the coefficient of variation (CV) and the intraclass correlation coefficient (ICC) are typically used to analyze the repeatability of measurements. Meanwhile, to calculate the change scores, the random error of measurement, the standard error of measurement (SEM), and the MDC (also known as the smallest detectable difference) are commonly used.

In this study, the SEM or “error” is not a mistake or mismatch in the usual sense, but rather an estimate made—with a high degree of confidence—of the range of values of the obtained measurements that can be expected when the test is performed again without changes to the sample or measurement conditions, i.e., the discrepancy between the observed and true score. The SEM is estimated as SEM = SD × $\sqrt{\left(1-\mathrm{ICC}\right)}$, where the SD is the pooled standard deviation of the test-retest assessments and the ICC is the coefficient of reliability.

The MDC represents the smallest change in a score, which is likely to reflect the true change, rather than the measurement error alone. It is calculated as
$$\mathrm{MDC}\;=\;Z-\mathrm{score}\;(\mathrm{CI})\;\times \;\mathrm{SEM}\;\times \;\sqrt{2}\;\;\mathrm{or}\;\mathrm{MDC}\;=\;1.96\;\times \;\mathrm{SEM}\;\times \;\sqrt{2}$$

The Z-score is 1.96, which corresponds to a z-score with a 95% confidence interval (CI) and a square root of 2 to adjust for sampling using two different measurements. The expression usually reflects the CI used, for example, for 95% of the interval, and is expressed as MDC95.

The purpose of this study was to determine the intra-rater minimum level of detectable change in the arm volume using circumference measurements of the limbs of individual patients, both segmentally and globally. The calculation first required determination of the intra-observer repeatability and the SEM.

## 2. Material and Methods

### 2.1. Study Design and Participants

#### 2.1.1. Design

A cross-sectional observational study of repeated measurements was conducted, and the second measurement was made blindly, i.e., without access to the value of the first measurement.

#### 2.1.2. Ethical Approval

This cross-sectional study was approved by the Human Research Ethics Committee of the University of Sonora (DMCS/CBIDMCS/D-50). All participants provided written informed consent after having the nature and intent of the study fully explained to them.

#### 2.1.3. Participants

A total of 25 women who had received a unilateral mastectomy for BC stage II or III participated in this study. The eligibility criteria were unilateral total mastectomy surgery at least nine months prior and an arm length from the wrist to the armpit of at least 40 cm. The exclusion criteria included no mastectomy, bilateral breast surgery, current upper-extremity infection, or lymphangitis. The aim of this work was not to compare arms, but to compare the two measurements for each arm segment; therefore, the analyses were conducted using 550 pairs of measurements (two measurements of each arm of 25 patients, 50 upper limbs, with 11 measurements per arm).

A sample size (i.e., number of arms) was calculated using G*Power 3.1.9.7 (Düsseldorf, Germany) [18], based on a desired confidence coefficient of 0.90, as described in previous publications on the reliability of limb volume determination using tape measures, and on a power of 0.90 and an alpha value of 0.05 [19]. For two testing sessions, a minimum sample size of 44 was required [20].

### 2.2. Measurements and Methods

#### 2.2.1. Arm Perimeter Measurements

One of the team’s researchers, with previous experience in the technique, carried out the measurement of the circumference of both arms. The measurements were conducted twice, with an interval of 15 min, and this timeframe was chosen to minimize the risk of true fluctuations in the arm volume between measurements.

The participants were seated with shoulder forward flexion, with the arm abducted at 30° and in supination and the elbow extended to approximately 180°, supported in a relaxed manner on a table. From the center of the distal wrist joint crease, 11 marks were made on the skin every 4 cm to an area near the armpit using a non-permanent skin marker pen, which were easily removed after the measurement. To measure the circumference of both arms, a retractable tape with a narrow blade (6 mm Lufkin W606PM) was used just above the marks. At the indicated level, the tape was wrapped around the arm perpendicular to the major axis of the limb, applying only the minimum pressure necessary for the blade tape to rest on the skin without causing indents. Afterward, the marks were completely erased using cotton wool moistened with physiological serum without reddening of the skin. This most often resulted in 11 circumference measurements covering 40 cm of the arm from the wrist to the axilla; thus, 10 volume segments were considered in this study (Figure 1).

#### 2.2.2. Arm Volume Measurements

The volume of each segment was calculated following the truncated cone model (circular cone frustum or frustum), (available as Appendix A).

Keeping in mind that the distance between the skin marks (g) does not correspond to the height of the cone (h) but, rather, the cone generator, the height of the cone (h) was determined according to Pythagoras’ theorem: “The square of the hypotenuse is equal to the sum of the squares of the other two sides” (Figure 2).

### 2.3. Statistical Analysis

The data are presented as the means ± standard deviations (SDs) and ranges. The data normality was assessed using the Shapiro–Wilk test.

To determine the confidence limits as measures of absolute reliability, the mean CV from individual test–retest CVs was used, and the Bland–Altman method was used for visual evaluation of the reliability of measurements and the agreement limits of arm volume.

Repeatability refers to the closeness of the agreement between successive readings obtained by the same method for the same material and under the same conditions (i.e., same operator, same apparatus, same setting, and same time). This was calculated by determining the ICC estimates and their 95% CIs based on two-way random effects, absolute agreement, and single rater measurement (ICC_2,1_) [21].

The absolute reliability was evaluated using the SEM, and the MDC_95_ was calculated both absolutely and as a percentage.

The statistical significance level was set at 5%, and all data were analyzed using SPSS statistical package version 23 (IBM, Armonk, NY, USA).

## 3. Results

Table 1 shows the main characteristics of the participants.

As reported in Table 2, the perimeter of the different arm segments increased from the wrist to the area near the armpit, where the volume was practically double that of the wrist (15.57 ± 0.99 vs. 30.69 ± 4.39). The consistency between the measurements of the different perimeters was very high, with ICCs above 0.994 (between 0.988 and 0.999) and a CV between the repetitions of the measurements ranging between 0.005 and 0.009.

The SEM was small, ranging in the different perimeters between 0.108 and 0.305 cm. The absolute SEM along the arm ranged from 0.3 to 0.8 cm, expressed in percentage terms ranging from 2.25% to 3.91% of the perimeter value.

As can be seen from Table 3, the volume of the different arm segments was calculated from the perimeter, and the volume of the different segments increased toward the axillary area, with average values ranging from 79.1 ± 11.51 to 273.57 ± 83.17 mL. The sum of the volumes of the 10 segments was calculated as 1794.8 ± 489.6 mL. The consistency (absolute agreement) between the determinations of the segmental volumes, calculated using different repeated measurements, was high, with the ICC ranging from 0.990 to 0.999 and a CV of the volume of the different segments varying between 0.7% and 1.7%. The variation between the total volumes calculated from the arm measurements was 0.07%.

The SEM in the different segments varied by 0.96–5.26 mL, and was 15.48 mL for the total arm volume. Meanwhile, the MDC in the volume at the segmental level ranged from 2.7 to 14.6 mL, or 3.37% to 7.57% if expressed as a percentage, and was 2.39% at the overall arm level.

## 4. Discussion

There is a high incidence of BC in women, as well as a high frequency of developing lymphedema after treatment. This makes the use of reliable, reproducible, and accurate methods for lymphedema evaluation even more necessary for both the diagnosis and follow-up of survivors. There are a variety of methods for the diagnosis of lymphedema, but the determination of perimeters is certainly the most commonly used in healthcare settings, as its results are known to correlate very well with those of more complex techniques [12,13,14]. The various methodologies used for the calculation of arm volumes from the measurement of perimeters differ in terms of the anatomical references used as the point of measurement, in addition to the overall length of segments from which the determination is made, but they have an apparent uniformity in considering the segments of the arm as a frustum (ref sear) and in using the following calculation for the volume of each segment: $$V=\frac{h}{12\Pi }+C^{2}+c^{2}+\left(C\;\times \;c\right)$$
(where *h* is the height of the cone, *C* is the greater perimeter, and *c* is the smaller perimeter).

However, a minor error can result from perimeter determination being generally applied by considering the distance on the skin, between the measurement points, to be the height of the cone when it is, in fact, the generator (with the value for the height of a cone being less than that of its generator) [22]. This error is reduced as the arm segment becomes more cylindrical and, conversely, is increased when the distance between the measurements or the segments becomes more cone-like in shape, with a greater difference between the generator and height values (Figure 2).

Intra-rater reliability evaluates repeatability, and the ICC_2,1_ of the circumference measurements indicates very good reliability along the different measured sections, varying between 0.988 and 0.999 (0.0994 ± 0.004), which are similar values to those published in numerous studies [15,23,24,25,26,27]. The reliability of the calculated segmental volumes, which ranged from 0.990 to 0.999 (0.0994 ± 0.003), is also similar [24,28,29,30,31,32]. This high reliability is one of the reasons why a recent study tried to answer the question of which method is best for determining excess arm volume. This study concluded that the calculation of the volume based on arm circumferences is the best measurement method for evaluating excessive arm volume over time [33]. In spite of the good interrater reliability that is usually presented in such studies, reliability is usually better if the patients are evaluated by the same therapist each time, i.e., the intra-rater reliability is superior to the interrater reliability [24].

Studies on the repeatability of arm circumference measurements are not uncommon. However, for correct follow-up of these patients, in addition to qualitatively assessing the repeatability (i.e., average, good, or high), it is also necessary to know what the random error of the method is and, equally, to bear in mind what the MDC is in order to be able to clinically contextualize the changes in measurements over time.

It is important to understand the degree of precision and the SEM of the instruments or methods of evaluation, both in the field of research and in the diagnosis or monitoring of patients, since this allows us to know the level of uncertainty of the clinical interpretation of the obtained data. The SEM can be expressed in absolute terms, and our data show that the SEM for different arm perimeter measurements varied between 0.108 and 0.305 cm. However, if we want to compare this SEM among people or populations with different heights or weights, it is preferable to express it as a percentage, and in our study, this ranged from 0.62% to 1.41% of the total perimeter, which is slightly higher than that found by [23]. For example, Chen et al. found that it varies between 0.5% and 0.7%, although in their study, they only conducted three measurements (i.e., forearm, shoulder, and upper arm), similarly to Devoogdt et al. [29], who reported 1.4%.

There are different criteria for establishing the diagnosis of lymphedema, one of which is that 2 cm of the difference of any segment of the limb confirms the diagnosis [34], but it is important to relativize this cut-off point, bearing in mind the SEM of this technique.

The volume is calculated from the perimeters, and in our work, the volume calculated for each of the segments presented an SEM ranging from 0.910 to 5.26 mL. In percentage terms, this is a variation of 1.22–2.73%. In related studies, it is uncommon to find information about the SEM of the segmental volume, although lymphedema is not always widespread throughout the limb and may instead be located in a particular part of the limb. In our study, the SEM of the entire limb (taken as 40 cm from the distal wrist crease) was 15.4 mL (0.86%). Of note, it has been observed that the greater the distance between the perimeter measurement points, the greater the SEM [27,29].

If a difference between the volume of the extremities, or segments, greater than 5% is to be used as a diagnostic criterion for lymphedema [17], it would be useful to take into account the SEM of the volume calculation in order to assess the limitations of the diagnostic decision.

The MDC is a calculation derived from the SEM and the Z-score of the ICC of the repeatability and is an easily interpreted and very useful measure in the follow-up of patients, since it reflects the minimum change that has to occur in the patient such that the diagnostic method can be used for detection with a high degree of reliability, i.e., the minimum amount of change that is unlikely to be due to an unintended variation in measurement [35]. In our study, the MDC varied between 3.4% and 7.6% at the segmental level, and these differences in the MDC between segments are probably due to the fact that the three-dimensional configuration of the different limb portions is not uniform.

In our study the MDC for the entire limb was 2.39%, which, in our sample, corresponds to 42.9 mL; this is below the 3.5% referred to by Devoogdt et al. [29] and the 7.5% referred to by Taylor et al. [36].

The limits of the volumetric agreement between the two measures, estimated from the measurement of the arm perimeters, are graphically depicted in the Bland-Altman plot in Figure 3.

The agreement between the measures are uniform regardless of the limb volume size, within 84.3 mL (−59.3 to 35.5), with a 95% confidence level.

This study highlights the importance of healthcare professionals following up on BC patients in order to understand the MDC when applying upper extremity circumference measurements for volume determination in the context of the diagnostic reliability of lymphedema. In addition, it emphasizes the need for the determination of not only the entire upper limb volume but also its various segments.

While the objective of this study was the determination of detectable MDC in the determination of arm volume and its segments, and not the comparison of the MDC between arms with/without lymphedema, the main limitation of the present study was that the data pool of all arms was analyzed, regardless of whether or not the arms had lymphedema.

Future studies could establish a comparison of the MDC between arms with/without lymphedema.

## 5. Conclusions

The MDC in the volume of the upper limb varies in the different segments due to the non-uniform three-dimensional configuration of the different sectors of the arm. An MDC of 2.39% for the volume of the upper limb, even though it might be small, should be kept in mind for a more accurate interpretation of the differences in the volume between arms or of the changes in the volume obtained in the follow-up of BC survivors.

## Figures and Tables

**Figure 1 healthcare-08-00285-f001:**
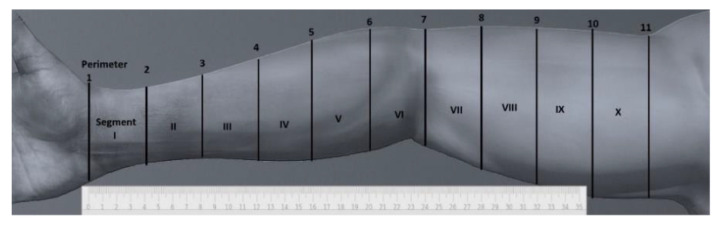
Locations of the perimeter measurements and the resulting segments.

**Figure 2 healthcare-08-00285-f002:**
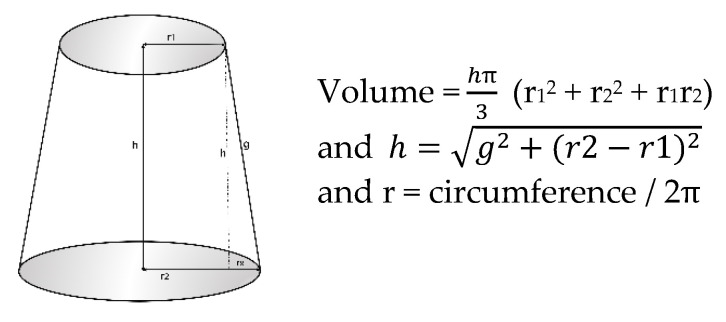
Basic components of a cone frustum.

**Figure 3 healthcare-08-00285-f003:**
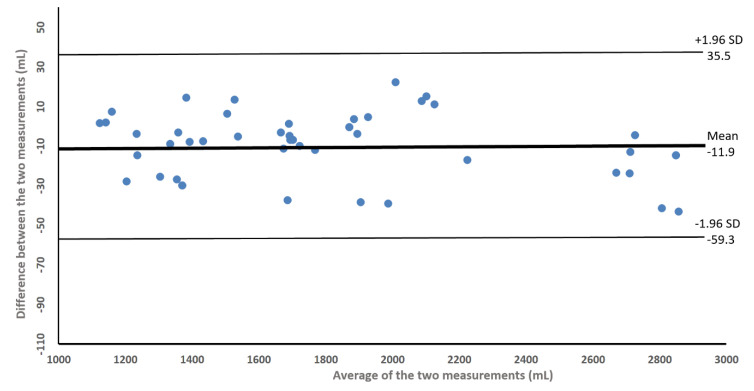
Bland–Altman plot showing the limits of agreement for the test–retest data on arm volumes.

**Table 1 healthcare-08-00285-t001:** Characteristics of the participants.

Affected side (right/left) (*n*)	11/14		
Lymphedema (yes/no) (*n*)	5/20		
	**Mean**	**SD**	**Range**
Age (years)	53.6 ± 10.7		(79–39)
Years since surgery	5.8 ± 4,0		(15–2)
BMI (kg/m^2^)	27.8 ± 4.3		(36.6–20.3)
Diff. % between arm volumes in women without lymphedema (*n* = 20)	3.9 ± 2.5		(0.5–7.9)
Diff. % between arm volumes in women with lymphedema (*n* = 5)	44 ± 22.7		(11.1–60.7)

**Table 2 healthcare-08-00285-t002:** Values and reliability of the perimeter measurements of the upper limb.

Perimeter	Mean	SD	ICC	Confidence Interval 95%			CV	SD	Min.	Max.	SEM	SEM%	MCD	MCD%
Perimeter 1a	15.57	0.99	0.988	(0.978	−	0.933)	0.006	0.004	0.000	0.018	0.108	0.69	0.3	1.92
Perimeter 1b	15.58	1.01												
Perimeter 2a	16.90	2.28	0.989	(0.981	−	0.994)	0.008	0.010	0.000	0.052	0.239	1.41	0.7	3.91
Perimeter 2b	16.87	2.27												
Perimeter 3a	19.10	2.76	0.994	(0.989	−	0.996)	0.009	0.008	0.000	0.037	0.214	1.12	0.6	3.11
Perimeter 3b	19.04	2.85												
Perimeter 4a	22.16	3.05	0.990	(0.982	−	0.994)	0.009	0.011	0.000	0.059	0.305	1.38	0.8	3.82
Perimeter 4b	22.10	3.10												
Perimeter 5a	24.41	2.99	0.995	(0.992	−	0.997)	0.006	0.006	0.000	0.031	0.212	0.87	0.6	2.40
Perimeter 5b	24.31	2.97												
Perimeter 6a	25.30	2.90	0.996	(0.993	−	0.998)	0.006	0.004	0.000	0.026	0.183	0.72	0.5	2.01
Perimeter 6b	25.21	2.91												
Perimeter 7a	25.54	3.25	0.996	(0.993	−	0.998)	0.005	0.005	0.000	0.031	0.206	0.80	0.6	2.23
Perimeter 7b	25.50	3.14												
Perimeter 8a	26.80	3.73	0.997	(0.995	−	0.998)	0.006	0.004	0.000	0.021	0.204	0.76	0.6	2.11
Perimeter 8b	26.73	3.70												
Perimeter 9a	28.37	3.95	0.998	(0.996	−	0.999)	0.005	0.004	0.000	0.021	0.177	0.62	0.5	1.73
Perimeter 9b	28.26	3.98												
Perimeter 10a	29.73	4.17	0.997	(0.994	−	0.998)	0.006	0.005	0.000	0.025	0.229	0.77	0.6	2.13
Perimeter 10b	29.63	4.15												
Perimeter 11a	30.69	4.39	0.999	(0.997	−	0.999)	0.005	0.003	0.000	0.015	0.139	0.452	0.4	1.25
Perimeter 11b	30.66	4.46												

SD, standard deviation; ICC, intraclass correlation coefficient; CV, coefficient of variation; Min., minimum value; Max., maximum value; SEM, standard error of measurement (absolute values); SEM%, SEM percentage values; MDC, minimal detectable change (absolute values); MDC%, MDC percentage values.

**Table 3 healthcare-08-00285-t003:** Values and reliability of the volume estimation of the upper limb.

Segment	Mean	SD	ICC	Confidence Interval 95%	CV	SD	Min.	Max.	SEM	SEM%	MCD	MCD%
Segment Ia	79.16	11.51	0.993	(0.987 − 0.996)	0.010	0.007	0.000	0.028	0.963	1.22	2.7	3.37
Segment Ib	79.07	11.84										
Segment IIa	95.45	25.78	0.990	(0.983 − 0.994)	0.015	0.018	0.002	0.093	2.578	2.70	7.1	7.49
Segment IIb	95.03	25.67										
Segment IIIa	121.07	33.52	0.994	(0.990 − 0.997)	0.017	0.014	0.001	0.069	2.596	2.14	7.2	5.94
Segment IIIb	120.45	34.60										
Segment IVa	159.91	43.69	0.990	(0.983 − 0.994)	0.016	0.020	0.000	0.110	4.369	2.73	12.1	7.57
Segment IVb	159.21	44.29										
Segment Va	191.82	48.26	0.995	(0.991 − 0.997)	0.012	0.013	0.000	0.062	3.412	1.78	9.5	4.93
Segment Vb	190.21	47.62										
Segment VIa	200.16	43.44	0.991	(0.984 − 0.995)	0.010	0.009	0.000	0.057	4.121	2.06	11.4	5.71
Segment VIb	198.85	42.90										
Segment VIIa	208.83	51.85	0.993	(0.988 − 0.996)	0.011	0.001	0.000	0.075	4.338	2.08	12.0	5.76
Segment VIIb	207.89	49.22										
Segment VIIIa	230.81	63.82	0.997	(0.995 − 0.998)	0.012	0.008	0.000	0.040	3.495	1.51	9.7	4.20
Segment VIIIb	229.54	62.83										
Segment IXa	257.60	70.55	0.997	(0.994 − 0.998)	0.011	0.009	0.000	0.042	3.864	1.50	10.7	4.16
Segment IXb	256.20	71.79										
Segment Xa	273.57	83.17	0.996	(0.993 − 0.998)	0.012	0.012	0.000	0.061	5.260	1.92	14.6	5.33
Segment Xb	270.68	81.52										
Total arm a	1794.8	489.6	0.999	(0.997 − 0.999)	0.007	0.008	0.000	0.035	15.483	0.863	42.9	2.39
Total arm b	1782.9	485.7										

SD, standard deviation; ICC, intraclass correlation coefficient; CV, coefficient of variation; Min., minimum value; Max., maximum value; SEM, standard error of measurement (absolute values); SEM%, SEM percentage values; MDC, minimal detectable change (absolute values); MDC%, MDC percentage values.

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
