# Peer review of "Determination of the Minimum Detectable Change in the Total and Segmental Volumes of the Upper Limb, Evaluated by Perimeter Measurements"

_healthcare, 2020, doi:10.3390/healthcare8030285_

Round 1
Reviewer 1 Report
The objective of this study was to determine the intrarater minimal detectable change (MDC) in the arm volume (globally and segmentally) by using circumference measurements. The same rater have determined the 11 perimeters of each arm at 4cm intervals from distal crease of the wrist in the directions of the armpit. The authors have reported the MDC ranges from 3.37%-7.57% at segmental level and 2.39% at the global level of the arm. It is interesting study but the manuscript could not be considered for publication in the current for following concerns. Major concerns: 1) There are few studies which have reported the reliability and reproducibility and the use of minimal detectable change (MDC) in the arm volume by using circumference measurements. (It may not be the purpose of this study but there are also studies which have used this approach to evaluate the outcome of lymphoedema treatment as well). Could authors please elaborate the novelty of the approach used in the study? 2) The major limitation of this study is very small sample size. The authors have stated that the major objective of this study was to determine the detectable minimal detectable change (MDC) in the arm volume (globally and segmentally) by using circumference measurements but not to compare the MDC between arms with/without lymphedema. Under this context, increasing the sample size would increase significance of the study statistically and also enhances robustness of the experimental design. Minor concerns: 1) Please replace "CB" with "BC" if the authors intention was to mention breast cancer (BC) on Ln # 53 of page # 2. 2) Could authors please use consistent approach to refer to Lymphedema. The authors have used Lymphedema (Example: Ln # 52) and Lymphoedema (example: Ln # 55) across the manuscript. Could authors please use either one of them throughout the manuscript. 3) Please replace "between arms volume women without" with "between arms volume in women without" in Table 1. 4) Please replace "between arms volume women witht" with "between arms volume in women with" in Table 1. 5) Please correct the typos in the x-axis and y-axis labels for Figure 3.6) Could authors please proofread the manuscript for syntax errors.
Author Response
Response to Reviewer 1 Comments
We appreciate your quick evaluation of the manuscript and the corrections you've pointed out to us. Thank you very much.
We have corrected the errors that you have pointed out, we have sent a second time to the English Edition service of the publisher to correct the syntactic errors; and we answer the queries that you have regarding what the value of the article is and in relation to the size of the sample.
We hope that we are responding to the remarks you have kindly made.
Again, and sincerely, thank you very much.
Major:
Point 1: There are few studies which have reported the reliability and reproducibility and the use of minimal detectable change (MDC) in the arm volume by using circumference measurements. (It may not be the purpose of this study but there are also studies which have used this approach to evaluate the outcome of lymphoedema treatment as well). Could authors please elaborate the novelty of the approach used in the study?.
Response 1: The objective of the study was to determine global and segmental MDC of the method, when it is performed by the same assessor on the same person in similar measurement conditions. In fact, there are few studies that analyze the minimum detectable change (MDC) and particularly few of the circumference measurement method, despite being the most used method in the world for the scraninng of the secondary lymphedema and follow up of the lymphedema in survivors to BC, and that is a practical contribution of our work. Repeatability studies are often limited to the calculation of the ICC and based on the ICC qualify the repeatability of the method as poor, good, very good or excellent. But they do not calculate the MDC, which should be known by the caregivers of these patients in order to interpret with a greater degree of certainty the result of the measurement of the total or segmental arm volume.
First we must know the general MDC of the method, and then analyze the MDC of the technique applied to different clinical situations, since not all arms with lymphedema present it in the same area of the limb, nor with the same degree, nor with the same consistency.
We believe, as some reviewers do, that this is a strength and not a weakness of our work.
Point 2: The major limitation of this study is very small sample size. The authors have stated that the major objective of this study was to determine the detectable minimal detectable change (MDC) in the arm volume (globally and segmentally) by using circumference measurements but not to compare the MDC between arms with/without lymphedema. Under this context, increasing the sample size would increase significance of the study statistically and also enhances robustness of the experimental design.
Response 2: Although we understand the reviewer's precautions, we must keep in mind that there are recognized methods used in the scientific literature to determine sample size based on expected variation and desired statistical power, (MDC is a form of repeatability analysis). Our sample was not a convenience sample, but we previously calculated the required size, based on data from other published studies and with the parameters commonly used in this type of study. Applying these criteria (intrarater, intra-subject study and a single measurement method), the resulting sample size is 44 pairs of samples, (44 upper limbs), and we have used 50.
By way of example, and without going into detail, we show some repetitiveness studies of this method and other techniques that have nothing to do with the technique used by us, to underline the size of the sample they have used, making it clear that the size of our sample is not inappropriate:
- https://pubmed.ncbi.nlm.nih.gov/32043125/, sample size: 30
- https://pubmed.ncbi.nlm.nih.gov/16445334/ (Compare between three groups) and the sample size of each of the three groups was: 19, 22 and 25.
Repeatability studies of other techniques:
- https://pubmed.ncbi.nlm.nih.gov/30676238/ (comparing the variation of two techniques) size sample: 30.
- https://pubmed.ncbi.nlm.nih.gov/30374253/ (balance) sample size: 22
- https://pubmed.ncbi.nlm.nih.gov/31044694/ (in two groups, sick and healthy): sample size: 20 and 20.
Minor:
Point 1: Please replace "CB" with "BC" if the authors intention was to mention breast cancer (BC) on Ln # 53 of page # 2
Response 1: We have corrected, by replacing in text "CB" by "BC"
Point 2: Could authors please use consistent approach to refer to Lymphedema. The authors have used Lymphedema (Example: Ln # 52) and Lymphoedema (example: Ln # 55) across the manuscript. Could authors please use either one of them throughout the manuscript.
Response 2: We have unified in the text the term lymphedema. We have not done so in the bibliography, since both concepts “lymphoedema” and “lymphedema” are correct and used in scientific publications, so we have kept the one used in the original articles we have consulted and used to elaborate our manuscript.
Point 3: Please replace "between arms volume women without" with "between arms volume in women without" in Table 1.
Response 3: We have corrected, by replacing in Table 1.
Point 4: (is 3, repeated)
Response 4: Responded to in answer 3.
Point 5: Please correct the typos in the x-axis and y-axis labels for Figure 3
Response 5: We have corrected in Figure 3
Point 6: Could authors please proofread the manuscript for syntax errors.
Response 6: We have sent for the second time to the Editorial English Correction Service, for correction. (English Editing Articles Status: Id.: 21387. Invoice englhis-21387)
Reviewer 2 Report
In this manuscript, authors trying to determine the intrarater MDC in the volume of the upper limb both segmentally and globally, using circumference measurements for the evaluation of upper limb volume of breast cancer survivors. MDC at the segmental level ranged from 3.37% to 7.57% and was 2.39% at the global level of the arm and these results are based on 25 (small sample size) adult women who had received a unilateral mastectomy for breast cancer stage II or III involved in this study. My comments are very positive. The manuscript contains interesting and quality of the latest literatures were presented in the manuscript is good. Finally, I believe that it is a good candidate for being published in journal. I have small suggestion to author to include patients ethnicity.
Overall Rating: good quality but of interest to a more specialized readership
Author Response
We appreciate your evaluation, about the quality and the practical interest of the manuscript, we also appreciate your quick review.
Round 2
Reviewer 1 Report
The objective of this study was to determine the intra-rater minimal detectable change (MDC) in the arm volume (globally and segmentally) by using circumference measurements. The same rater have determined the 11 perimeters of each arm at 4cm intervals from distal crease of the wrist in the directions of the armpit. The authors have reported the MDC ranges from 3.37%-7.57% at segmental level and 2.39% at the global level of the arm. The authors have addressed the comments and can be considered for publication.